# Propagated Circulating Tumor Cells Uncover the Potential Role of NFκB, EMT, and TGFβ Signaling Pathways and *COP1* in Metastasis

**DOI:** 10.3390/cancers15061831

**Published:** 2023-03-17

**Authors:** Jerry Xiao, Utsav Sharma, Abolfazl Arab, Sohit Miglani, Sonakshi Bhalla, Shravanthy Suguru, Robert Suter, Reetu Mukherji, Marc E. Lippman, Paula R. Pohlmann, Jay C. Zeck, John L. Marshall, Benjamin A. Weinberg, Aiwu Ruth He, Marcus S. Noel, Richard Schlegel, Hani Goodarzi, Seema Agarwal

**Affiliations:** 1School of Medicine, Georgetown University, Washington, DC 20057, USA; 2Department of Pathology, Center for Cell Reprogramming, Georgetown University, Washington, DC 20057, USA; 3Lombardi Cancer Center, Georgetown University, Washington, DC 20057, USA; 4Department of Biochemistry and Biophysics, University of California, San Francisco, CA 94158, USA; 5Department of Medicine, The Ruesch Center for the Cure of Gastrointestinal Cancers, Georgetown University Medical Center, Washington, DC 20057, USA; 6Department of Pathology, Georgetown University Medical Center, Washington, DC 20057, USA

**Keywords:** circulating tumor cells, CDX model, metastatic cancer, colon cancer, lung cancer, pancreatic cancer

## Abstract

**Simple Summary:**

Metastasis is the primary cause of cancer-related deaths, but is poorly understood. Circulating tumor cells (CTCs) seed distant sites are a promising model system for studying metastasis. Unfortunately, CTCs are very rare and there are few methods for efficiently establishing in vitro and in vivo CTC models. We extended our recently published method for routinely establishing CTC cultures from liquid biopsies (blood draws) of breast cancer patients to diverse cancers, i.e., colon, lung, and pancreatic. We also successfully established CTC-derived xenograft (CDX) models from the expanded CTCs. We used these models to identify genomic markers and pathways associated with metastases.

**Abstract:**

Circulating tumor cells (CTCs), a population of cancer cells that represent the seeds of metastatic nodules, are a promising model system for studying metastasis. However, the expansion of patient-derived CTCs ex vivo is challenging and dependent on the collection of high numbers of CTCs, which are ultra-rare. Here we report the development of a combined CTC and cultured CTC-derived xenograft (CDX) platform for expanding and studying patient-derived CTCs from metastatic colon, lung, and pancreatic cancers. The propagated CTCs yielded a highly aggressive population of cells that could be used to routinely and robustly establish primary tumors and metastatic lesions in CDXs. Differential gene analysis of the resultant CTC models emphasized a role for NF-κB, EMT, and TGFβ signaling as pan-cancer signaling pathways involved in metastasis. Furthermore, metastatic CTCs were identified through a prospective five-gene signature (*BCAR1*, *COL1A1*, *IGSF3*, *RRAD*, and *TFPI2*). Whole-exome sequencing of CDX models and metastases further identified mutations in constitutive photomorphogenesis protein 1 (*COP1*) as a potential driver of metastasis. These findings illustrate the utility of the combined patient-derived CTC model and provide a glimpse of the promise of CTCs in identifying drivers of cancer metastasis.

## 1. Introduction

As high as 90% of all cancer-associated deaths are directly attributable to the metastatic spread of cancer cells from the primary site to a secondary organ [1,2]. Emphasizing its important role, metastasis was listed as one of the six hallmarks of cancer, which describe the biological capabilities acquired during the multistep development of tumors [3]. However, despite its role in cancer, metastasis remains a mysterious phenomenon largely due to a lack of clinically robust and accessible individualized patient-derived models [4,5].

Previous patient-derived models for identifying drivers of metastasis have depended on solid tissue biopsies of matched primary and metastatic lesions [6,7]. While numerous prospective therapeutic compounds have successfully inhibited metastases in preclinical models, most have been unsuccessful in clinical studies [8], suggesting that these models do not faithfully replicate clinical metastasis. For instance, early studies evaluating the SRC inhibitors dasatinib and saracatinib in various cancer models (pancreatic, thyroid, prostate, ovarian, and melanoma) showed promise preclinically but failed to demonstrate efficacy in human trials [8,9,10]. Thus, the frequent failure of these and other promising candidates in preventing metastasis in clinical trials suggests that these preclinical models—which often rely on either cell-line-based or tumor explants—do not faithfully recapitulate clinical metastasis [8].

An evolving approach to understanding metastases is to leverage circulating tumor cells (CTCs [1]), which have detached from the primary tumor and/or from a metastatic site and entered the circulatory system, thereby undergoing the early stages of metastasis [11]. CTCs can be obtained from routine blood draws, subjecting patients to minimal discomfort in addition to providing easy access for serial monitoring of disease [11,12,13,14,15].

Importantly, recent advances have demonstrated that CTCs are a highly heterogeneous, information-rich, and distinct metastatically driven rare population of cancer cells that may provide in-depth insights into metastasis [12,13]. Simultaneously, it is not possible currently to fully capitalize on the considerable promise of CTCs due to challenges associated with identifying and obtaining CTCs in sufficient numbers to carry out mechanistic and functional studies [11]. The primary difficulty is that CTCs are ultra-rare, typically numbering <10 cells per mL of blood [16]. As a result, molecular and mechanistic studies that exploit CTCs require first expanding CTCs ex vivo [11]. Such methods fall into two broad categories, namely, in vitro cell culture and in vivo CTC-derived xenografts (CDXs) [17,18]. Unfortunately, these methods are also dependent on the ability to isolate relatively large quantities of CTCs to initiate cultures. On average, only 10–25% of all initiated CTC cultures result in successful cultures [11,17,19,20,21,22,23]. Similarly, the formation of patient-derived CDXs has been difficult to achieve when extracted CTC counts are lower than 400 CTCs/7.5 mL blood [18,24,25,26]. Thus, CTC cultures and CDX models have been difficult to establish on most patients.

Once successfully propagated, mechanistic studies of patient-derived CTC cultures and CDXs have already contributed to key efforts in understanding metastatic phenomena [11]. For instance, transcriptomic profiling of breast-cancer-derived CTCs reaffirmed the importance of mTORC1, TGF-β, and p53 signaling in metastasis [27]. Another report demonstrated that CTCs from a patient diagnosed with estrogen receptor (ER) positive bilateral breast carcinoma and bone metastases could be cultured without estrogen supplementation, emphasizing a clinically relevant phenomenon of the decoupling of estrogen signaling and cell survival [28]. The treatment of CDX models has also been shown to accurately mirror the response of the patient to therapies [29]. For instance, CDX models derived from patients diagnosed with small cell lung cancer showed reduction in tumors following cisplatin and etoposide treatment, consistent with the response of the patients from whom the tumors were derived [30].

By contrast, we recently published a novel method using a density-dependent, non-biased methodology (i.e., without the use of antibodies) for selecting and expanding CTCs and associated CD45^+^ cells in cultures from patients with metastatic breast cancer [27]. Following our report, others have also confirmed the presence of co-inhabiting CD45^+^ cells in successful short-term pancreatic CTC cultures [31]. Here we report the creation of patient-derived CTC cultures and CDX models from patients diagnosed with colon, lung, and pancreatic cancer. Notably, these CTC-derived CDX models were characterized by highly aggressive, de-differentiated primary tumors with an ability to form macro-metastatic lesions similar to their corresponding patient tumors. Pathway and differential gene expression analyses comparing patient-matched whole blood, cultured CTCs, and corresponding CDX models also highlighted the potential cancer-agnostic role for NF-κB, EMT, and TGFβ signaling pathways crucial to metastasis. Furthermore, differential gene analysis of metastatic CTCs pared down alterations in RNA expression among CTCs to reveal five candidate genes whose expression correlated with poorer prognosis in the TCGA pan-cancer dataset. Finally, whole-exome sequencing comparing mutations between formalin-fixed paraffin-embedded (FFPE) tissue biopsies, cultured CTCs, and CDX primary tumors identified both mutational concordance/discordance as well as a prospective driver, *COP1*, in cancer metastasis. Taken together, the novel in vitro and in vivo models established in this study have contributed to the identification of key molecular signatures associated with cancer metastasis.

## 2. Results

### 2.1. CTCs Can Be Established from Multiple Cancer Types

Previously, we reported the successful initiation of short-term cultures of breast-cancer-derived CTCs using a novel methodology [27]. Here we demonstrate that this platform can be applied to additional cancer types, namely, lung, colon, and pancreatic cancers. These cancers were selected as they are among the most common cancers diagnosed in the United States and together accounted for over 270,000 deaths in 2021 [32]. To begin, we obtained 25 liquid biopsies from 21 consented individual patients and processed these samples for the expansion of CTCs as described previously (see Ref. [27] and Figure 1) [27]. Short-term cultures with viable cells were established from 5/7 colon, 5/5 lung, and 13/13 pancreatic cancers, resulting in an overall success rate of 92.0% for establishing CTC cultures. Thus far, CTC cultures derived from lung cancers were observed to survive the longest in culture (59 ± 27 days), followed by colon (56 ± 25 days) and pancreatic (39 ± 20 days) cancers. qRT-PCR amplification confirmed the expression of the CTC markers Cytokeratin 18 (CK18) and Vimentin (VIM) in all cultures, with an exception of CTC21, where no CK18 was detected (Appendix A).

### 2.2. CDXs Can Be Established from CTCs Expanded In Vitro

An important criterion for successfully establishing CDX models from liquid biopsies has usually been a high initial extracted CTC count (≥400 CTCs) [11,18,33,34,35] with one recent exception wherein recently, a CDX model from a non-small lung cancer was established using only 35 EpCAM^+^CD45^−^ cells [36]. Unfortunately, patients with a high CTC count in peripheral blood are extremely rare as usually <10 cells/mL are present in the patient’s blood. We therefore hypothesized that patient-derived CDX models could be more routinely established following an initial expansion phase in in vitro CTC cultures. Four patients with high-growth CTC cultures derived from lung (patient #19), colon (patients #21 and 22), and pancreatic (patient #26) adenocarcinomas were selected as candidates for establishing CDX models (Figure 2a). A detailed clinical summary for each patient is provided in the Appendix A.

To establish CDX models, CTCs separated from whole peripheral blood were first cultured for 28 days as described in Figure 2a. CTCs expanded in culture (5000 to 25,000 cells) were injected subcutaneously into the flanks of immunodeficient mice. These mice are referred to in this report as “first-generation CDXs.” All first-generation CDXs developed palpable primary tumors within 32 ± 6 days post-injection (*n* = 7 mice) and were euthanized by day 78 (mean 65 ± 30 days) (Figure 2b). Tumor tissues from first-generation CDXs were divided and reimplanted into another mouse of the same immunodeficient strain to generate the “second-generation CDXs.” The second-generation recipient mice developed palpable primary tumors at the site of the injection within 31 ± 10 days (*n* = 16 mice) and were euthanized by day 165 post-transplantation (mean 75 ± 29 days) (Figure 2b). Among all first- and second-generation CDX models, macro-metastatic nodules visible to the naked eye were observed in 21 of the 23 models on autopsy, involving an average of two distant organs per mouse (Figure 2c). Overall, the spleen was the most common site of metastasis (19/21 90.5%), followed by the axillary lymph nodes (14/21, 66.7%), lungs (7/21, 33.3%), liver (6/21, 28.6%), kidneys (1/21, 4.8%), and pancreas (1/21, 4.8%) (Figure 2c). Hematoxylin-eosin staining of tumors and macro-metastases indicated poorly differentiated carcinoma cells (Figure 2d). IHC staining using human-specific antibodies targeting GAPDH and VIM as well as immunofluorescent (IF) staining for EpCAM all demonstrated positivity in both CDX tumors and metastatic nodules. No detectable staining of human CD45 was observed in CDX tissues (Figure 2d).

Recently, cases of spontaneous lymphomagenesis in immunocompromised mice following the xenograft of human cells have been reported [37]. Therefore, we sought to dispel this possibility in the CDX mouse models. Following the isolation of RNA from CDX tissues and RNA sequencing, we used the CIBERSORT software [38], ProteinAtlas [39], and reactome pathway analysis to identify the proportions of human immune cells, comparing RNA obtained from the whole blood of the patient and the corresponding tumor tissue of the CDX model. CIBERSORT showed that CDX tissues were significantly depleted for not only CD4^+^ naïve and CD8^+^ T cells but also B cells, natural killer (NK) cells, and monocytes (Appendix A). Immune-cell-specific signatures derived from the ProteinAtlas [39] also indicated a decrease in the expression of genes specific for B and T cells, NK cells, dendritic cells, macrophages, plasma cells, and granulocytes (Appendix A). Finally, reactome pathway analysis demonstrated the depletion of human genes associated with the innate immune system in cultured CTCs and CDX tissues relative to samples obtained from healthy donors (Appendix A). Taken together, these findings rule out the presence of murine spontaneous lymphomagenesis.

### 2.3. CTCs and CDXs Express Transcriptomic Heterogeneity

To evaluate transcriptomic similarities or differences between healthy donor whole blood (HD), matched patient-specific whole blood (WBM), cultured CTCs (TC), and CDX models, bulk RNA-sequencing was performed on all samples, and differential gene analysis was carried out (Figure 3a). For analysis, HD and WBM samples were pooled together and compared against TC and CDX tissues. Compared with HD and WBM, 848 genes were significantly (*p*-value < 0.05) up-regulated and 21 genes were significantly down-regulated in TC and CDX models. When TC models were compared with WBM, 1828 and 1165 genes were significantly up- or down-regulated, respectively (Appendix A). Similarly, 3129 and 3419 genes were significantly up- or down-regulated, respectively, in CDX tissues compared with WBM (Appendix A). Finally, 1836 and 2731 genes were significantly up- or down-regulated in CDX tissues compared with TC (Appendix A). These data summarize pooled data for all four patients; patient-specific comparisons are further provided in Appendix A.

Previous studies including ours have identified activation of the epithelial-mesenchymal transition program as a critical event in the transition of a cancer cell from primary tumor to metastasis [31]. We therefore evaluated the expression of a panel of epithelial (KRT5, 7, 8, 18, and CDH1), mesenchymal (VIM, FN1, SNAI1, TWIST1, COL1A1, and PRRX1), and cancer stem cell (ALDH1A2, ALDH7A1, CD44, and CCND1) markers in all samples. We observed increased expression of markers associated with metastases in CTC-derived models (TCs and CDXs) compared with HD and WBM (Figure 3b).

Last, patient-specific differential gene analysis was performed comparing the TCs and CDX tissues with HD and WBM samples (Figure 3c–f). Notable genes that were up-regulated in patient-derived CTC models include GDA, FOXA2, GJB1, and NR2F2 (Appendix A). On the other hand, XIST and RPS4Y1 were the most down-regulated genes in each respective patient sample (Appendix A).

### 2.4. Five Genes Are Associated with High-Risk CTCs

Recent studies have indicated that CTCs exhibit considerable transcriptional heterogeneity [13]. A robust gene biomarker across multiple cancer types for CTCs has yet to be identified [40,41]. Therefore, we interrogated our data to identify genes whose expression was at least eightfold higher in TCs or CDX tissues compared with the matched WBM. Based on this criterion, 199 (Figure 4a and Appendix A) and 35 (Figure 4b and Appendix A) genes were identified from cultured CTCs and CDX tissues, respectively. Of these, we identified eight genes (*IGSF3*, *EMX1*, *TFPI2*, *CCL22*, *BCAR1*, *RRAD*, *CCL1*, and *COL1A1*) (Figure 4c) that were up-regulated in all CTC cultures and CDX tissues. Kaplan–Meier survival analyses using TCGA pan-cancer atlas clinical data and gene expression data of these eight genes further narrowed the signature to a subset of five genes (*BCAR1*, *COL1A1*, *IGSF3*, *RRAD*, and *TFPI2*) whose increased expression was significantly associated with poorer disease-free survival (Figure 4d and Appendix A).

In addition to a gene signature for CTCs, we also sought to determine if our CDX models could be used to identify prospective organ-specific metastatic signatures. For instance, transcriptomic signatures for lung and brain-specific breast cancer have been identified through the evaluation of matched xenografted tumors and resultant metastatic lesions [42,43]. Gene expression was compared between the CDX primary tumors and their respective lymph node, spleen, lung, and liver metastases. Across organs, 47, 10, 12, and 255 genes were uniquely expressed in the CDX spleen, lymph node, liver, and lung that were not overexpressed in the primary CDX tumor (Figure 4e–h). To further narrow the prospective organ-specific signatures, we set a threshold expression of greater than eightfold in the metastases compared with the CDX primary tumor. In all, four, two, six, and five genes met this criterion in spleen, lymph node, liver, and lung metastases, respectively, resulting in the final identification of four prospective organ-specific signatures (Figure 4i–l).

### 2.5. NF-κB, EMT and TGFβ as Putative Pathways for CTCs

Another method of evaluating transcriptomics involves the differential analysis of gene sets and signaling pathways through the Kyoto Encyclopedia of Genes and Genomes (KEGG) [44] and Gene Set Enrichment Analysis (GSEA) [45]. A patient-specific analysis comparing signaling pathways enriched in cultured CTCs or CDX tissues compared with WBM was performed using KEGG. Cultured CTCs and CDXs from the four patients showed enrichment of the following signaling pathways, among others: patient #19, mTOR1, NOD-like receptor, and NF-κB (Figure 5a); patient #21, TNF, PI3K-AKT, IL-17, and NF-κB (Figure 5b); patient #22, Chemokine, p53, NF-κB, IL-17, and TLR (Figure 5c); and patient #26, PI3K-Akt, TNF, and NF-κB (Figure 5d).

Notably, CTCs are composed of a highly heterogeneous population of cells [46,47]. While pathway analysis of cultured CTCs and CDXs revealed enrichment of shared signaling pathways, it was not surprising that differences in enriched signaling pathways were also observed, which can likely be attributed to the microenvironment in which the CTCs are present. For instance, PD-L1 and mTOR1 signaling were enriched in cultured CTCs from patient #19 but not the derivative CDX tissues (Figure 5a). Conversely, HIF-1 and TNF signaling were enriched in CDX tissues derived from patient #19 but not the source cultured CTCs. Similarly, other individually enriched signaling pathways were observed in either CDX tissues (e.g., chemokine signaling in CDX tissue from patient #21) or cultured CTCs (PDL-1 signaling and p53 signaling in CTC from patient #26) only (Figure 5b–d).

GSEA pathway analysis offers a more focused approach to cancer, evaluating 50 sets of genes associated with specific biological states or processes with minimal redundancy [48]. As described above, we compared the gene expression from TCs and CDX tissues with WBM from the same patient. The gene set that defines the epithelial-mesenchymal transition and TGFβ were significantly enriched in all four patients. Similarly, TNFα signaling via NF-κB was also enriched in models derived from all four patients. EMT, TGFβ, and NF-κB signaling are routinely identified in studies of cancer progression and metastasis [49] (Figure 5e–h and Appendix A). Other gene sets that were routinely enriched in CTC-derived models were G2M targets. These results are consistent with the findings from the KEGG analyses.

Furthermore, a functional network was generated through the input of the 199 gene prospective TC signature from Appendix A into String-DB, using the whole human genome as the background. Terms found to be enriched across the CTC gene signature were selected, and member proteins overlapping those processes and the CTC signature were plotted. Nodes represent all proteins produced by a single, protein-coding gene locus, either by associated Gene Ontology (GO), KEGG pathway, or WikiPathway (WP) terms. All enriched terms showed an FDR < 0.05 (Appendix A). Notably, core nodes within this signature involved the PI3K-Akt, MAPK, and NF-κB signaling pathways. Together, these data suggest NFκB, EMT, and TGFβ as putative pathways driving metastasis.

### 2.6. CDXs Can Be Queried for Metastasis-Driving Mutations

In addition to transcriptomic heterogeneity, CTCs are also known to acquire de novo mutations separate from their respective primary tumors [50,51]. In some cases, these de novo mutations have been identified as potential drivers of resistance to therapeutics [24,52]. Additionally, multiple de novo CTC-exclusive mutations have been identified in genes that are associated with human cancers, namely, *KRAS*, *BRAF*, and *PIK3CA* [53,54].

To evaluate the prevalence of mutational concordance/discordance in our samples, we performed whole-exome sequencing (WES). WES analysis of HD, patient-specific FFPE, TCs, CDX primary tumors, and corresponding distant metastatic CDX lesions was performed and compared against pathology reports from primary tumor biopsies obtained from patients where available (patients #21, 22, and 26). In patient #21, of the seven single-nucleotide polymorphisms (SNPs) reported in the primary tumor, five were identified in the matched TC, CDX, and CDX metastases (Table 1). In patient #22, a SNP in the cancer tumor suppressor gene *TP53* reported in the patient’s primary tumor was identified in the corresponding FFPE and CTC culture, but not the CDX primary tumor or distant metastases (Table 1). Finally, in patient #26, a SNP in *SF3B1* reported with an 11% variant frequency in the patient’s primary tumor biopsy was identified in the matched CTC culture, CDX, and CDX metastases (Table 1).

Next, a mutant allele fraction (MAF) was calculated to determine the presence of true SNPs within our samples (Figure 6a). Unfortunately, DNA sequencing of patient’s FFPE tissue samples resulted in poor-quality reads across a large majority of the genome, representing difficulties in whole-exome sequencing of patient tumors (Figure 6a). Despite these limitations, across all four samples, ≥80% of all FFPE identified mutations were recovered in corresponding TC and CDX models, indicating strong genetic fidelity across models (Figure 6a). On the other hand, some variability in mutations across samples did exist (Figure 6b–e). These findings ultimately confirm genetic fidelity across the two distinct model systems (TC and CDX) as well as demonstrate mutational heterogeneity consistent with the known behaviors of CTCs.

Next, we hypothesized that de novo oncogenic mutations identified in cultured CTCs or CDX primary tumors could signal a role in metastasis. The OncoKB database, curated from worldwide sources, provides detailed information regarding specific alterations in 682 cancer genes [55]. In 2021, the FDA recognized a portion of the OncoKB as a source of valid scientific evidence for level 2 (clinical significance) and level 3 (potential clinical significance) biomarkers. This meant that submissions to the FDA could use these data to support the clinical validity of tumor profiling tests. In patient #19, fifteen oncogenic mutations were CDX tumor-exclusive, affecting genes such as *CDK8*, *COP1*, and *MAP2K2* (Appendix A). In patient #21, mutated genes in cultured CTCs included *AKT1*, *SERPINB3*, *EGFR*, and *NOTCH4*, while *COP1* was once again mutated in CDX tumors (Appendix A). In CDX tumors derived from patient #22, *COP1* and *GNAS* were mutated, while *NUF2* was the only oncogenic mutation identified in CTC cultures (Appendix A). Multiple de novo mutations in *COP1*, *CDK8*, and *AKT2* were observed in CDX primary tumors derived from patient #26, while mutations in *SERPINB3*, *MGAM*, and *HDAC7* were observed in corresponding patient-matched cultures (Appendix A). In summary, mutation in *COP1* was identified in all four CDXs from diverse tumor types suggesting *COP1* as a potential driver of metastasis.

Mutations in specific genes could also be responsible for organ-specific patterns of metastasis [48]. We therefore sought to determine whether a consistent pattern of mutations driving metastasis could be identified based on specific organs (Figure 7a). No WES analysis was conducted on distant organs from CDX models derived from patient #21 due to poor-quality sequencing. When considering all CDX primary tumors, 60.4% of all identified mutations were unique to a single patient-derived primary tumor, while approximately 5.6% of all mutations occurred in all four sequenced primary tumors (Figure 7b).

To identify potential organ-specific metastatic mutations/genes, we developed an analytical workflow that is depicted in Figure 7c. Across all unique mutations in CDX spleens, 19.1% of mutations were shared across all three spleen samples queried (Figure 7d). Similarly, 4.5% and 13.5% of all unique mutations were shared across all three CDX lymph node and liver samples, respectively (Figure 7e,f). After filtering for only those genes that were included within the OncoKB database, we identified 60 unique genes, including *MTOR*, *NOTCH2*, *JAK1*, *JUN*, and *HIF1A*, that harbored mutations only within the liver samples and were not present within lymph nodes or spleen samples (Figure 7g and Appendix A). On the other hand, five genes (*KDR*, *RUNX1T1*, *RPS6KB2*, *NOTCH3*, and *EPOR*) exhibited mutations exclusively in CDX lymph node tissues. Finally, 17 genes exhibited mutational exclusivity in CDX spleens, including *YAP1*, *HDAC7*, *MAPK3*, and *SOX9* (Figure 7g and Appendix A).

## 3. Discussion

Circulating tumor cells could be key to understanding the metastatic cascade and developing directed therapies against metastasis. This study introduces a novel platform for isolating and culturing CTCs ex vivo that significantly improves upon previous attempts. Leveraging an unbiased density-dependent isolation process that is independent of selection by antibodies or microfluidic devices, we have established viable CTC cultures from 23/25 total samples spanning lung, colon, and pancreatic adenocarcinomas (Figure 1), confirming the high yield previously described by us with breast cancer clinical samples [27]. In addition, the expanded CTC cultures were successfully used here to generate robust CDX models with extensive macro-metastases.

Previous attempts at culturing CTCs ex vivo have been hampered by the requirement of high numbers of CTCs isolated directly from whole blood [11,17,18]. Historically, antibody-dependent isolation methods have been used to concentrate CTC [15,56]. However, in most patients, CTCs are ultra-rare (<10 cells/mL) [15]. For instance, the isolation and culture of CTCs from patients diagnosed with gastroesophageal cancers using the commercially available RosetteSep^®^ could only establish CTC cultures from two out of 41 (a 4.9% success rate) patients, both of whom had at minimum 109 CTCs/7.5 mL of blood [22]. Such a high CTC count is rare in even the most metastatic cancer patients. In addition, most previous reports of CTC cultures have been targeted toward the culture of CTCs derived from a single cancer type [11]. In contrast to previous studies, in this report we demonstrate the successful expansion of cultured CTCs from regular peripheral blood draws in patients diagnosed with one of three distinct tumor types (colon, lung, and pancreatic adenocarcinomas) using a single generalized culture technique (Figure 2a). This culture system enables the harvest of CTCs along with CTC-supporting cells that may be present within whole blood [27]. In this study, vimentin expression at the cell surface was used in combination with CK18 and/or EpCAM to define the CTCs as these cells undergo EMT. However, CD45^+^ cells also express vimentin, but it is only present intracellularly [57], while CTCs express vimentin at the cell surface as shown in Figure 2d. 

Similarly, in previous attempts to establish CDX models, only those mice infused with samples from patients with very high CTC counts (≥400) successfully established CDXs [18,27,33,34,35]. Given the rarity of CTCs, seeking patients with high numbers of CTCs in the blood is not a viable strategy for the routine initiation of CDX models. Our ability to successfully expand CTCs in cultures with a high success rate (92%) offered a plausible strategy to routinely initiate CDX models using ≧5000 CTCs/injection, opening the use of this technology to initiate both CTC short-term cultures and long-term CDXs in a large cohort of patients spanning across multiple cancer types. In the proof of concept described here, we used CTCs from four patients with diverse cancers that were first expanded through short-term cultures (14–28 days) to attain numbers suitable for the establishment of patient-derived CDXs (Figure 2). We not only successfully established CDX models using cultured CTCs from all four patients but also demonstrated replicable patterns of metastasis through multiple generations. Of the few CDX models previously described, many fail to demonstrate metastasis, only generating a tumor at the site of injection [25,34,35] with a high latency period of at least 6 months. The four CDX models generated here all exhibited macro-metastatic nodules in distant organs within 2–3 months. Our models thus recapitulate the physiological function of CTCs as a metastasis-initiating population. 

While our primary purpose in expanding CTCs ex vivo was to enhance the numbers of CTCs available for initiating CDX models, our data suggest that it is also plausible that the short-term culture selected for a highly aggressive and adaptable CTC population that was particularly suited to survival in adverse conditions such as ex vivo culture. For instance, cultured CTCs were enriched in expression for not only epithelial markers but also for mesenchymal and cancer stem cell markers (Figure 2, Figure 3 and Appendix A). This expression of EMT plasticity in CTCs is of particular interest as it has been demonstrated that mesenchymal-expressing CTCs are more likely to cluster and form metastatic lesions in vivo [12]. Furthermore, previous single-cell RNA sequencing of isolated CTCs demonstrated that CTCs are a highly dynamic and heterogeneous population [58]. Genomic heterogeneity seen among CDX tumors and their corresponding metastases suggests that at least some level of heterogeneity was maintained in the cultured CTCs and CDX models (Figure 6 and Figure 7 and Appendix A). Further studies plan to evaluate how culturing and expanding CTCs affects the composition of the heterogenic population at the single-cell level.

We leveraged the patient-derived CTC cultures and CDX models to determine the enrichment of specific molecular pathways and gene signatures in cultured CTCs and CDXs derived from them in a pan-cancer cohort. In a previous study, we identified the enrichment of TNFα signaling via NF-κB in CTC cultures derived from breast cancers [27]. In this study, TNFα signaling via NF-κB was once again a commonly enriched pathway in cultured CTCs and their corresponding CDX models derived from all four patients (Figure 5). Thus, the repeated identification of TNFα suggests that it is a critical and conserved pan-cancer pathway involved in metastasis. Other reports have also identified TNFα signaling in metastasis in a variety of cancers, identifying perturbations associated with enhanced metastatic potential [59]. In the clinical setting, anti-TNFα inhibitors are now entering clinical trials to treat cancers [60]. The indirect targeting of TNFα signaling through NF-κB inhibitors has also been more widely studied in cancers as NF-κB is frequently constitutively activated in most human cancers [61]. In the non-canonical pathway, NF-kB activates a signaling cascade that works through the TNF receptor to introduce a set of cytokines, eventually resulting in the activation of inflammation [61]. Several pharmacological strategies have been employed to target NF-κB in cancers, including the inhibition of upstream IKK phosphorylation, immunomodulatory agents such as thalidomide, or proteasome inhibitors such as bortezomib [61]. The existing evidence of the importance of TNFα in human cancers is therefore supplemented with the novel findings in the current study. Taken together, these studies strongly suggest that direct or indirect targeting of TNFα signaling could be a potentially potent strategy for developing metastasis-specific therapeutic agents. The routine establishment of individual patient-derived CTC cultures and matched CDX models will provide a key biological resource for screening molecules that target TNFα as specific anti-metastatic agents.

In this study, differential gene expression analysis revealed a prospective CTC signature consisting of five genes (*BCAR1*, *COL1A1*, *IGSF3*, *RRAD*, and *TFPI2*) whose expression (a) was robustly enriched across all CTC-derived samples (Figure 4c) and (b) was associated with poorer disease-free survival using the pan-cancer TCGA dataset (Figure 4d). Notably, all five genes that constitute the metastatic signature have been previously individually linked to metastatic phenomena.

BCAR1 has previously been identified as a novel binding partner of mutant TP53, ultimately promoting cancer cell invasion [62]. Collagen type I alpha 1 (*COL1A1*) is perhaps the most strongly associated with the development of metastases in cancer as a critical element of the tumor microenvironment [63,64]. IGSF3 acts as a promoter of hepatocellular carcinoma progression and metastasis through the activation of the NF-κB signaling pathway [65]. Knockdown of Ras-related associated with diabetes (RRAD), a GTPase commonly implicated in metabolic function and hepatocellular carcinomas, suppressed cancer cell invasion, proliferation, and EMT [66,67]. Finally, tissue factor pathway inhibitor 2 (TFPI2) has been shown to promote cancer metastasis through increased perivascular migration and ERK signaling [68,69].

Finally, we provide evidence of mutational discordance between patient primary tumors, FFPE, CTC cultures, and CDX tissues, demonstrating the de novo acquisition of mutations in oncogenic genes occurring during metastasis (Figure 6 and Figure 7). According to the classical viewpoint of cancer as a clonal disease, CTCs can develop an enhanced ability to metastasize due to the acquisition of novel genetic mutations [70]. While a large proportion of mutations were commonly shared between FFPE, CTC cultures, and CDX tissues, a focus on de novo mutations in cultured CTCs and CDX models could provide insight into prospective drivers of metastasis. For instance, mutations in constitutive photomorphogenesis protein 1 (*COP1*) were observed in all four CDX tumors evaluated (Figure 6 and Appendix A). In breast cancer cells, the depletion of COP1 resulted in the stabilization of the c-Jun protein and the subsequent enhancement of cancer cell growth/migration and in vivo metastasis [71]. The prevalence of *COP1* across all four models could suggest a possible selective pressure for inactivating *COP1* mutations in metastasis. Interestingly, several other genes whose up-regulation is commonly associated with increased metastatic potential also exhibited either culture or CDX model mutational exclusivity, including the serine-protease inhibitor *SERPINB3*, tyrosine kinase *EGFR*, and *AKT* isoforms 1 and 2 (Figure 6 and Appendix A). The latter is particularly interesting as the various Akt isoforms are known to have varied effects on metastatic potential [72]. These findings may be partially supported by our own findings, which identified *AKT1* (an isoform generally associated with cancer cell growth and proliferation) mutations in cultured CTCs and *AKT2* (an isoform more commonly associated with metastasis) mutations in CDX tumors [72]. Finally, reports of CDK8 [73], EGFR [74], MAP2K2 [75], and NOTCH signaling [76] and *NUF2* [77] mutations associated with enhanced metastasis are encouraging indications of the accuracy of the CTC-derived models established here. 

This study provides evidence that CTCs from diverse cancers can be efficiently expanded ex vivo. More importantly, we demonstrate that such cultured CTCs can, in turn, be used to reliably, rapidly, and routinely establish CDX models. We have furthermore leveraged individual patient-derived CTCs and CDX tissue to genetically characterize CTCs. Scaling up these technologies would allow the establishment of sufficient numbers of CTCs and CDX models from individual cancers to identify biomarkers and to screen existing and novel molecular entities for their capacity to target metastases. Patient-derived CTCs and CDX models could also be used to design personalized therapeutic regimens for individual patients. Therefore, the application of this novel platform to a broader cohort of patients could potentially lead to an improved understanding of the underlying mechanisms of metastasis as well as leverage CTCs as a key resource in developing a new generation of precision-based personalized cancer treatments and vaccines for cancer patients including metastatic cancers.

This study has a few limitations. We did not use germline mutation for each patient to identify all variants and, instead, focused only on the cancer-related genes. Thus, the study does not identify metastasis-specific mutation(s) that are not among previous datasets of cancer-related genes. In future studies, germline mutations will be used to identify metastasis-specific somatic mutation(s). Additionally, this study did not carry out single-cell RNA sequencing of CTC models generated to identify the degree of heterogeneity and tumor evolution present in CTC-derived models. Both of these will be part of the future studies currently being carried out in our laboratory.

## 4. Methods

### 4.1. Patient Enrollment

Patients were recruited, consented, and enrolled at the Medstar Georgetown University Hospital Medical Oncology clinics in compliance with the Health Insurance Portability and Accountability Act (HIPAA) and Georgetown University Institutional Review Board (IRB) procedures (approval ID: MODCR00001156), managed through the Survey, Recruitment, and Biospecimen Collection shared resource (SRBSR) of the Lombardi Comprehensive Cancer Center. All patients provided written informed consent for the study. Inclusion criteria for patient enrollment included adult (>18 years old), male or female patients with histologically confirmed metastatic disease and primary lung, colon, and pancreatic cancers, with untreated or treated metastatic disease, and with no limit of prior treatment lines. Exclusion criteria included any one or more of the following: (1) use of chemotherapy and/or antibody-based therapy less than a week before blood collection and (2) use of radiotherapy in the past 2 weeks unless there exist metastatic lesions beyond the radiated lesion. Healthy donors with no known health conditions at the time of consent were also enrolled. In all patients, two to four tubes of blood (~7–8 mL/tube) were drawn. When applicable and available, blood was drawn from existing IV ports; if blood draws required skin punctures, an initial waste tube was disposed prior to processing. Deidentified pathology reports and clinical annotation regarding patient tumors and history were provided by the SRBSR. Patient clinical characteristics are detailed in Appendix A.

### 4.2. CTC Isolation from Whole Blood Using FiColl-Paque

All patient peripheral blood samples were processed within 90 min of collection from patients. FiColl-Paque (Cytiva Life Sciences, Marlborough, MA, USA, cat. no. 17-440-02) based separation of CTCs was performed as previously reported [27]. Briefly, samples were mixed with 1× Hank’s Balanced Salt Solution (HBSS) at 1:1 volume/volume ratio with whole blood at room temperature. An amount of 6 mL of the HBSS:blood sample was then split evenly into two 15 mL tubes containing 3 mL FiColl-Paque each, being careful not to mix samples with FiColl-Paque. The remaining HBSS:blood volume was split evenly into two 50 mL tubes containing 15 mL FiColl-Paque each. The samples were processed as described previously [27]. Upon the completion of the final wash, cell pellets from 50 mL tubes were resuspended in 1 mL culture medium/tube. The cells were resuspended via gentle pipetting and plated for short-term cultures. Cells resulting from 3 mL Ficoll-Paque tubes were used to extract DNA and RNA designated as a whole blood match (WBM).

Blood processing for healthy donor blood proceeded in the same manner as the isolation from enrolled patients. In healthy donors, two tubes of 7.5 mL blood each were drawn and processed via FiColl-Paque. Following all centrifugation and wash steps, cell pellets for healthy donors were reserved for RNA/DNA extraction only.

### 4.3. CTC Cultures

The cells were resuspended in culture medium and plated for short-term cultures at 37 °C for 14+ days as described previously [27]. The cultures were supplemented with fresh medium every 3 days and washed every 6 days with 1× PBS by centrifugation at 400× *g* for 4 min at 4 °C. Every 6–10 days, cells were harvested via manual pipetting and resuspended in 1 mL of culture medium for automated trypan blue viability assays using the Thermo Scientific Invitrogen Countess II, Waltham, MA, USA, (AMQAX1000 cell counter). Phase microscopy images were taken using the EVOS FL Auto Imaging System (Thermo Fisher) at 20× zoom.

### 4.4. CTC-Derived Xenograft Models

All animal protocols and procedures for this study were approved by the Institutional Animal Care and Use Committee (IACUC) at Georgetown University (Protocol 2020-0033). Briefly, 5–6 weeks old female NOD.Cg-*prkdc^scid^-IL2rg^tm1Wjl^*/SzJ (NSG) mice were purchased from Jackson Laboratory (JAX stock #005557) and housed at the Georgetown University animal facility. Mice were housed in a standard 12 h light–dark cycle and fed standard mouse feed and water ad libitum. Prior to xenografts, all mice were acclimated in standard housing conditions for a minimum of one week after delivery. For CDX first-generation studies, cells were harvested, washed with 1× PBS, and pelleted via centrifugation at 300× *g* for 4 min. The cells were resuspended in a 1:1 volume/volume solution of 1× DPBS (Life Technologies) and growth factor reduced Matrigel (Corning) up to a final total volume of 100 µL/injection. For second-generation CDX transplantation, cryopreserved 1 mm^3^ tumor chunks in a 10% DMSO/90% FBS solution were thawed at room temperature and washed in 1× PBS twice prior to transplantation. Baseline weights were recorded for each animal and monitored throughout the study. The animals were anesthetized using 2–3% isoflurane and monitored for depth of anesthesia and respiration throughout the procedure. The surgical site was depilated and sterilized using ethanol and PVP Iodine prep swab sticks (PDI healthcare, cat. no. S41125) prior to xenografting. For first-generation CDXs, the cells were injected subcutaneously, and for second-generation CDXs, a shallow incision was made, and pieces of the first-generation CDX primary tumor no larger than 1 mm^3^ were implanted in the unilateral dorsal flank of the mice. Post-operative wound closure, pain management, and monitoring were performed as per the approved animal protocol. The animals were monitored weekly for the development of palpable tumors. Following palpable tumor formation, the animals were monitored twice weekly, and the tumor size was measured via calipers. The tumor volume was calculated by the following formula: volume = 0.5 × L × W^2^ (where L is the length/longer dimension and W is the width/shorter dimension). Tumor-bearing mice were euthanized as per the animal protocol at specific time points depending on the experimental model as described in results. In general, mice were euthanized when the tumor reached 1500 mm^3^, 9 months post-injection, or when the mice exhibited signs of poor health. Post-euthanasia, primary tumors, metastases, and other organs were immediately harvested and sectioned for immunohistochemistry (IHC), cryopreservation in 10% DMSO:90%FBS (volume/volume), and flash-freezing in liquid nitrogen for DNA/RNA extraction.

### 4.5. RNA/DNA Extraction

RNA and DNA were isolated immediately from whole blood samples (“WBM”), healthy donor whole blood (“HD”), patient FFPE tissue biopsy cores (“FFPE”), CTC cultures (“TC”), and CDX organs. All RNA and DNA extraction kits were used following the manufacturer’s protocols. All RNA samples were treated with RNAse-free DNAse provided by the manufacturer’s kits or with RNase-Free DNAse Set (Qiagen, cat. No. 79254). All DNA samples were treated with RNAse provided by the manufacturers to remove contaminant RNAs. FFPE tissue cores of patient primary tumors or metastases were collected by the SRBSR, sectioned at the HTSR, and collected in microcentrifuge tubes. RNA and DNA were extracted from FFPE tissues via a QIAamp DNA FFPE Tissue Kit (Qiagen, cat. no. 73504) or RNeasy FFPE Kit (Qiagen, cat. no. 56404). For TC samples, cultured cells were detached from culture plates by gently pipetting the plates with 1× PBS or via mechanical removal with a cell scraper. The cells were pelleted via centrifugation (400× *g* for 4 min at 4 °C) and washed one time with 1× PBS prior to RNA/DNA extraction. For TC samples, RNA was extracted using RNAqueous-Micro Total RNA Isolation Kits (Thermo Fisher, cat. no. AM1931) with added DNAse treatment. The resulting RNA was eluted in 13.5 µL of provided elution buffer. DNA from TC samples was extracted using Qiagen DNeasy Blood and Tissue Kits (Qiagen, cat. no. 69504). DNA was eluted in 40 µL of ddH2O.

For CDX tissues, flash-frozen tissue samples that had been prepared at the time of euthanasia were used. The tissues were first ground into a powder under liquid nitrogen freezing in mortar and pestles. RNA and DNA extraction proceeded immediately following the grinding of the CDX tissues. For DNA and RNA extraction, three kits were used: (1) Qiagen DNeasy Blood and Tissue Kits (Qiagen, cat. no. 69504) for DNA extraction and (2) RNeasy Mini Kit (Qiagen, cat. no. 74104) for RNA.

All DNA and RNA samples underwent an initial round of quality control via spectroscopy using a NanoDrop 2000 (Thermo Scientific). DNA samples were considered high quality with 260/280 ratios of ≥1.80. RNA samples were considered high quality with 260/280 ratios of ≥2.00. All DNA and RNA samples were immediately stored at −80 °C for long-term storage. Minimal freeze–thaw cycles were allowed to limit degradation.

### 4.6. Immunohistochemistry and Immunofluorescence

Tissues collected from the euthanized mice were fixed in formalin and paraffin embedded at the Histopathology & Tissue Shared Resource core at Georgetown University, American Histolabs Inc. at Gaithersburg, MD, USA, or the Yale Pathology Tissue Services at Yale University. Hematoxylin & Eosin of sectioned slides was performed at one of the three institutions, as well as at the Lombardi Comprehensive Cancer Center. Briefly, the tissue sections were deparaffinized by melting at 60 °C in an oven for 45 min followed by 2× xylene treatment for 20 min each. The slides were then rehydrated, and antigen retrieval was performed in the manufacturer’s specified buffer (citrate buffer (pH 6.0), EDTA buffer (pH 8.0), or Tris/EDTA buffer (pH 9.0) made in house) at 97 °C for 20 min in a PT module (Labvision, Kalamazoo, MI, USA). Endogenous peroxidase was blocked by using 0.3% hydrogen peroxide in methanol for 30 min in the dark followed by the incubation of the slides in a blocking buffer (0.3% bovine serum albumin in TBST (0.1 mol/L of TRIS-buffered saline (pH 7.0) containing 0.05% Tween-20)) for 30 min at room temperature. The slides were then incubated with the primary antibody diluted in blocking buffer overnight at 4 °C. The antibodies used in this study include anti-GAPDH (1:200 dilution, Abcam, cat. no. ab128915) and anti-CD45 clone 2B11 + PD7/26 (Ready-to-use, Agilent Dako, Santa Clara, CA, USA, cat. no. GA75161-2). After washing away the primary antibodies, the slides were incubated with the secondary antibody (goat anti-mouse conjugated to horseradish peroxidase, supplied in Mouse and Rabbit Specific HRP/DAB (ABC) Detection IHC kit from Abcam, cat. no. ab64264) to target either GAPDH or CD45 for one hour at room temperature. After washing, the slides were treated with DAB substrate (Abcam, cat. no. ab64238) per the manufacturer’s protocol. Immunohistochemistry for Vimentin was performed using the IHCeasy Vimentin Ready-To-Use IHC kit (ProteinTech, cat. no. KHC0039) following the manufacturer’s protocol or an in-house standardized operating procedure. Control slides using tissue from wild-type non-surgery NSG mice and human-specific tissue microarrays provided by the Yale Pathology Tissue Service were used to optimize the DAB staining exposure. The exposure times were selected such that clear staining could be observed in positive human control slides but not in uninjected mouse slides. No exposure lasted longer than 10 min. Following the DAB staining, the slides were washed two times with ddH2O and counterstained using hematoxylin solution (Abcam, cat. no. ab220365) or the counterstain supplied in the kit for 1–2 min. The slides were subsequently washed twice in water to remove the excess counterstain and covered using glass coverslips secured with Fluorsave Aqueous Mounting Medium (Sigma-Aldrich, St. Louis, MI, USA, cat. no. 345789). If coverslips needed to be removed, the slides were soaked overnight in ddH2O at 4 °C.

For EpCAM immunofluorescence, the slides were incubated with the primary antibody, anti-EpCAM (1:200 dilution, Abcam, cat. no. ab223582), diluted in blocking buffer overnight at 4 °C. After washing away the primary antibodies, the slides were incubated with the secondary antibody (Donkey-anti-mouse conjugated with Alexa Fluor 568, Invitrogen, cat. no. A21202) for one hour at room temperature. After incubation, the slides were washed 2× with TBST and 1× with TBS and incubated for 10 additional minutes using Hoescht 33342 Solution (Thermo Fisher, cat. no. 62249) per the manufacturer’s protocol. The slides were then washed 2× with TBST and 1× with TBS and mounted using either Fluorsave Aqueous Mounting Medium or ProLong Gold Antifade Reagent with DAPI (Thermo Fisher, cat. no. P36935).

### 4.7. Quantitative Real-Time PCR

Following RNA extraction, RNA was converted to cDNA using the LunaScript RT SuperMix Kit (NEB, cat. no. E3010) according to the manufacturer’s specifications. For all reactions, a minimum of 25 µg and maximum of 100 µg of RNA were used as input, with a final reaction volume of 20 µL. A Bio-Rad T100 thermal cycler was used to incubate the reactions using the following program: 25 °C for 2 min, 55 °C for 10 min, 95 °C for 1 min, and 4 °C hold. All cDNA reactions were subsequently diluted with ddH2O to a final concentration of 25 µg initial input RNA/20 µL. All cDNA were stored at −20 °C for long-term storage, with minimal freeze–thaw cycles allowed to limit degradation.

For quantitative RT-PCR, Luna Universal qPCR MasterMix (NEB, cat. no. M3003) was used according to the manufacturer’s specifications. Briefly, 20 µL reactions were mixed using 10 µL of Luna Universal qPCR Master Mix, 0.5 µL per Forward and Reverse Primer (diluted to 10 µM) (Appendix A), 1 µL of cDNA template, and nuclease-free water to the final reaction volume. PCR primers were provided in 5′ to 3′ format. All qRT-PCR reactions were run in a Bio-Rad CFX 96 well thermal cycler with Thermo Scientific PCR 96-well plates (Thermo Fisher, cat. no. AB-0800). All qRT-PCR reactions were analyzed using the 2^−ΔΔCt^ method. All qRT-PCR samples were run through gel electrophoresis using in-house-made 1.5% agarose gels run at 95V for 35 min prior to imaging. All electrophoresis images were taken using a GE AI600 RGB Gel imaging system.

### 4.8. Next-Generation RNA and Whole-Exome Sequencing Library Preparation and Sequencing

All next-generation sequencing library preparation and sequencing was performed with the support of established third-party commercial companies. Library preparation and RNA-sequencing were conducted by Psomagen (Rockville, MD, USA). Libraries were created using TruSeq stranded mRNA sample preparation, and paired-end reads with a targeted read length of 151 bp were performed on a NovaSeq 6000 S4 sequencing platform. Low-input RNA-sequencing was performed, requiring a minimum of 50 ng of RNA for processing. For each sample, 60M total minimum reads were targeted; for CDX samples with the potential for mouse cell contamination, 100M total reads were targeted. For whole-exome sequencing, libraries were created using SureSelect V5-post library preparation kits, and paired-end reads with a targeted read length of 151 bp were sequenced on a NovaSeq 6000 S4 sequencing platform. A minimum of 500 ng of high-quality DNA was required for library preparation and sequencing. For each sample, 100× raw depth coverage was targeted. Only one sample per CTC model was used for sequencing.

### 4.9. Bulk RNA-Sequencing Bioinformatics Analysis

Paired-end read files were trimmed using Trimmomatic [78] and aligned to the human transcriptome (GRCh38.p13) reference from Gencode v33 using salmon v0.14. The data quality was verified using FastQC. The salmon output was processed in R using the tximport and tidyverse packages [79]. The differential expression analysis was performed in R using DESeq2 [80]. Normalized feature counts were used with the gene set enrichment analysis (GSEA) software and Molecular Signature Database (MSigDB), available at http://www.broad.mit.edu/gsea/ (accessed on 12 December 2022) [45]. GSEA was performed in R using the open-source package fgsea (version 1.20.1) [81]. Gene sets for fgsea input were downloaded from the MSigDB database [45]. KEGG analysis was performed in R using the open-source package ClusterProfiler (Release 4.2.1) [82]. Other R source packages used for figure development include but are not limited to ggsci and gplots as well as Matlab R2021a/b. CIBERSORT was accessed via https://www.cibersort.stanford.edu/ (accessed on 18 December 2022), and non-normalized feature counts were input following the developer’s instructions [38].

### 4.10. Whole-Exome-Sequencing Bioinformatics Analysis

To identify the mutations and genetic alterations associated with metastatic phenotypes, whole-exome sequencing of cultured CTCs and corresponding patient FFPE tissue and healthy donor tissue was performed. Briefly, deep sequencing and exome capture was performed using the SureSelect V5-post and Illumina platform with 200× coverage. Using BWA (Burrows-Wheeler Aligner, v0.7.17), we indexed human (hg38) and mouse (mm10) chromosomal sequences (fasta format). Then BWA mem2 module was used to align the raw FASTQ files to the human reference genome (GRCh38.p13). The contaminating mouse sequences were removed using the XenofilteR package in R. The package estimates edited the distance to classify sequences as mouse or human and then removed the mouse sequences.

The resulting bam (mapped to hg38 or xeno-filtered) files were analyzed using GATK and SnpEff, which generated the predicted SNP and InDel events. This information was used for the final interpretations in this manuscript. For reproducibility purposes, we are maintaining our WES scripts in the following repository: https://github.com/goodarzilab/WES. Mutant-allele tumor heterogeneity (MATH) scores were generated for CTC models and compared with the patient’s FFPE. Somatic mutations, consisting of point mutations, insertions, and deletions across the exome, were identified using the VariantDx custom software [83]. Somatic mutations were annotated against the set of mutations in the COSMIC (v.84) database. Revel scores for missense mutations were calculated to determine the mutation potential as cancer drivers by CHASMplus [84]. CTC or FFPE exclusive mutations were defined based on an MAF ≥ 0.5 and ≤0.25 in the opposing corresponding sample type from the same patient. Shared mutations were defined based on an MAF ≥ 0.5 in all samples. Only SNVs with a mutation count ≥ 4 were considered real and included within the differential enrichment analysis.

## 5. Conclusions

Circulating tumor cells (CTCs) are a unique subpopulation of cancer cells predisposed to metastasis. However, previous attempts to study CTCs and identify prospective drivers of metastasis have been hindered by an inability to obtain numbers of CTCs from an individual patient sufficient for study and for use in clinical decision making. In this report, we take advantage of combined in vitro tissue cultures and in vivo mouse-derived xenograft models to propagate CTCs derived from metastatic colon, lung, and pancreatic cancers. Furthermore, we demonstrate a potential workflow for extracting clinically relevant information using propagated CTCs. Differential gene analysis of resultant CTC models implicated increased activation of NF-κB and TGFβ signaling pathways in the CTC metastatic phenotype. Gene expression analysis across all three cancer types subsequently found a common increase in the expression of five genes (*BCAR1*, *COL1A1*, *IGSF3*, *RRAD*, and *TFPI2*), which may further inform biomarker study for identifying CTCs. Additionally, whole-exome sequencing of CDX models and metastases further identified the presence of a common mutation in *COP1*, implying an important role for this mutation in metastatic spread. Ultimately, this report serves as a blueprint that future large-scale studies evaluating CTCs can use to ultimately identify prospective drivers of cancer metastasis.

## Figures and Tables

**Figure 1 cancers-15-01831-f001:**
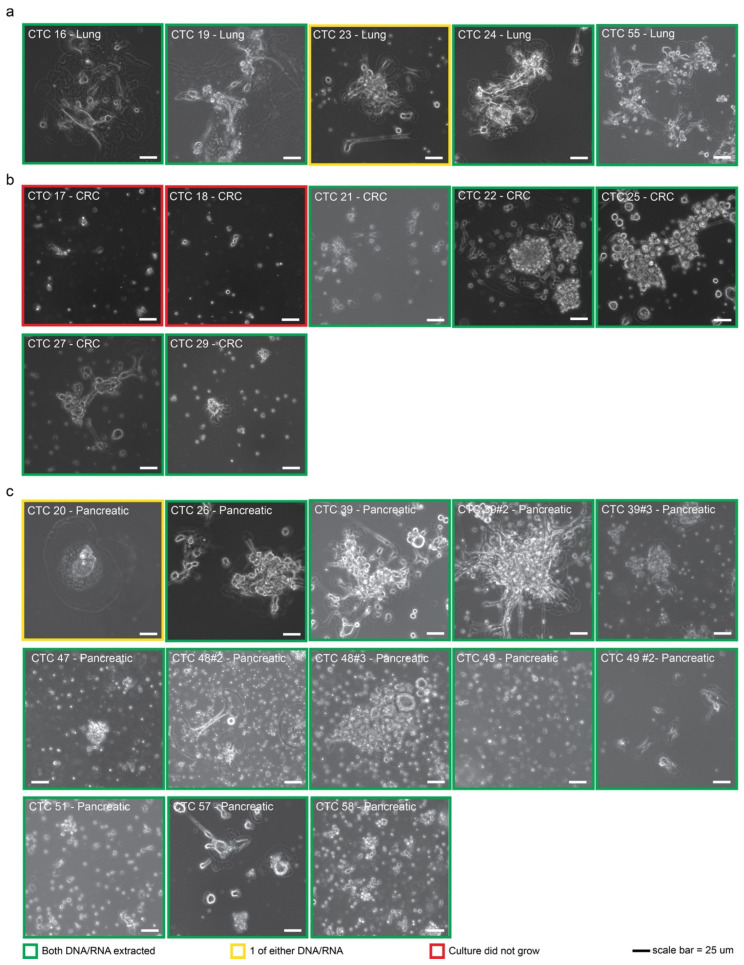
CTCs derived from multiple cancer types can be cultured. Representative phase images of CTC cultures derived from (**a**) lung, (**b**) colon (CRC), and (**c**) pancreatic metastatic tumors are shown. Images are outlined based on the ability to gather high-quality RNA or genomic DNA. Scale bar = 25 um.

**Figure 2 cancers-15-01831-f002:**
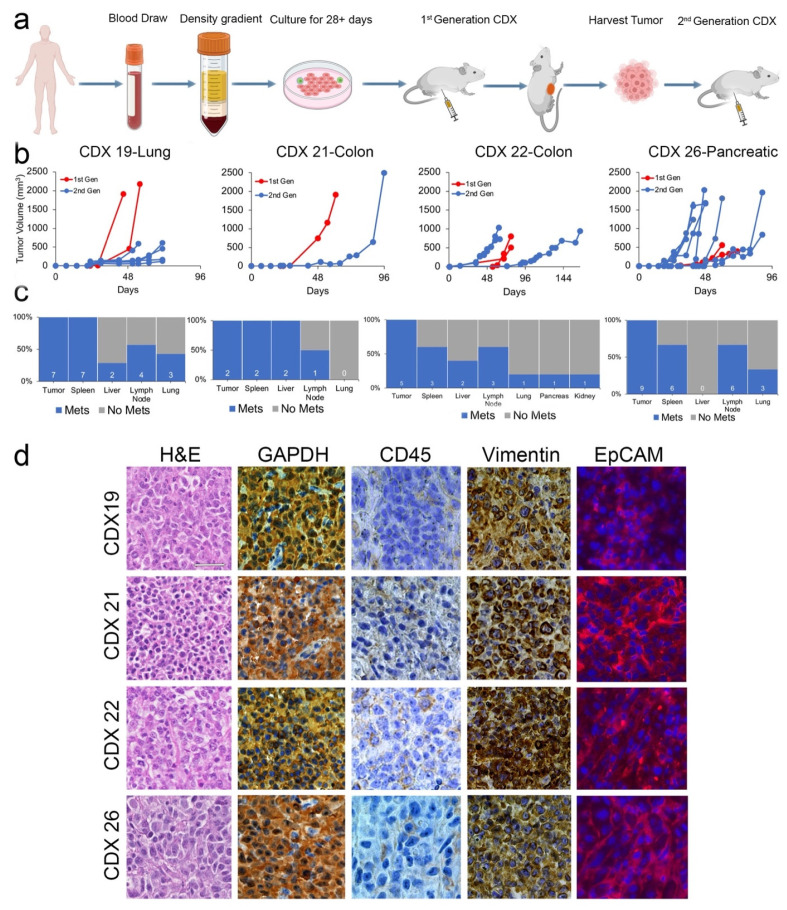
Four patient-derived CDX models were established following a period of short-term CTC culture. (**a**) Schema of CDX model formation created using Biorender.com. (**b**) Growth charts indicating the progression of CDX tumors. The first-generation CDX models were established via the injection of cultured CTCs; the second-generation CDX models were established via the serial re-engraftment of tissue derived from the first-generation CDX primary tumors. (**c**) Corresponding bar graphs indicating the number of CDX models with evidence of tumors in distant organs, separated by patient. (**d**) Representative hematoxylin & eosin (H&E, column 1), DAB chromogen staining of human-specific GAPDH (column 2), common leukocyte marker CD45 (column 3), and vimentin (column 4), and immunofluorescence staining for EpCAM was performed on first-generation CDX primary tumors. Scale bar = 25 um. CDX: CTC-derived xenograft.

**Figure 3 cancers-15-01831-f003:**
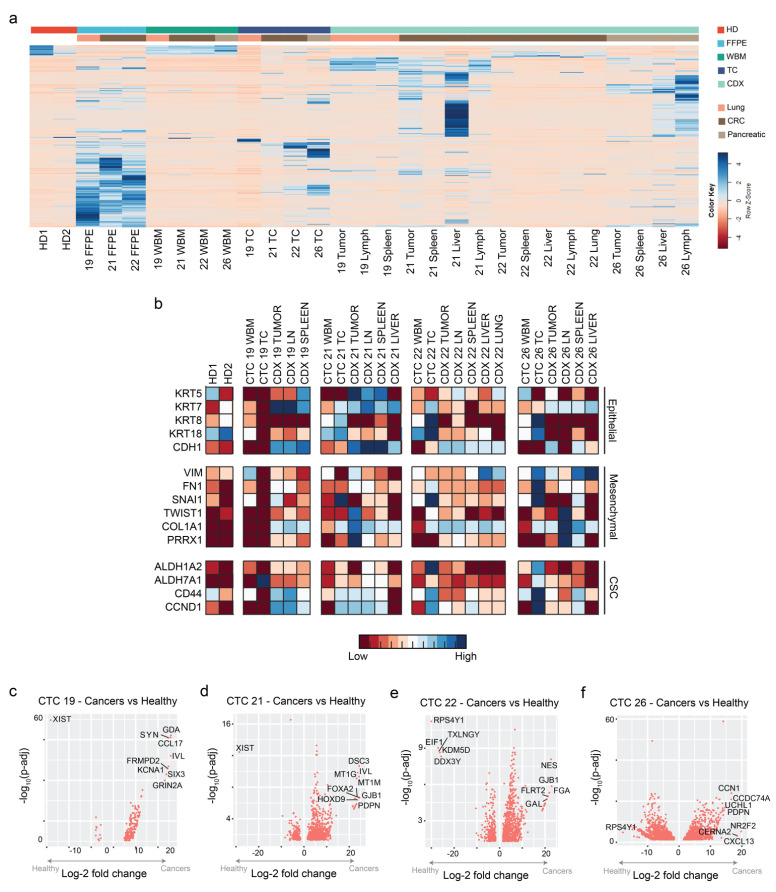
Bulk RNA-sequencing on cultured CTCs and CDX tissues shows transcriptomic heterogeneity. (**a**) A clustergram depicting mRNA expression in the various samples evaluated. (**b**) A panel of epithelial (KRT5, KRT7, KRT8, CDH1, and EpCAM), mesenchymal (VIM, FN1, SNAI1, TWIST1, COL1A1, and PRRX1), and cancer stem cells (ALDH1A2, ALDH7A1, CD44, and CCND1) was extracted. In general, CTC-derived models demonstrated increased expression relative to healthy donor and whole blood samples. (**c**–**f**) Volcano plots for all significantly differentially expressed genes between patient-specific cancer and healthy samples. A select few of the most statistically significant differentially expressed genes are labeled for each patient. Each dot plotted represents a unique differentially expressed gene identified. HD: healthy donor; WBM: whole blood match, TC: CTC culture; FFPE: formalin fixed paraffin embedded; and CDX: cultured CTC-derived xenograft.

**Figure 4 cancers-15-01831-f004:**
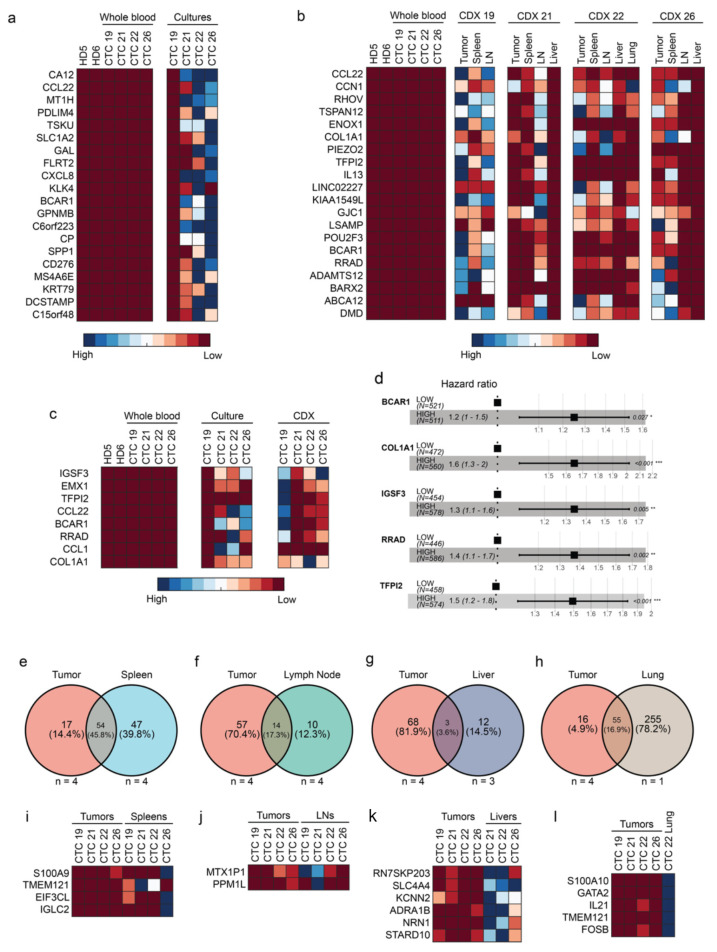
A prospective CTC-specific signature was identified. Differential gene analysis was used to determine a 35 gene signature (top 20 most differentially expressed depicted here) (**a**) and a 199 gene signature (**b**) that could reliably identify any CTC cultures or CDX tissues, respectively. (**c**) A final gene signature composed of eight genes (IGSF3, EMX1, TFPI2, CCL22, BCAR1, RRAD, CCL1, and COL1A1) that were routinely overexpressed in any CTC-derived sample was identified. (**d**) COX proportional hazards ratios were calculated based on gene expression of the eight CTC-signature genes and overall survival within the relevant TCGA pan-cancer atlas databases. Higher expression of five of the eight genes was associated with a poorer overall survival. (**e**–**h**) Differential gene analysis was performed on a CDX organ-specific level relative to a common control of whole blood samples. Venn diagrams depict the amount of overlap in genes that were overexpressed in tumors and genes that were overexpressed in the various respective distant organs. Organ-specific metastatic signatures were then identified by comparing the expression of genes in CDX tumor samples and respective distant organs. Ultimately, (**i**) a four-gene spleen metastatic signature, (**j**) a two-gene lymph node metastatic signature, (**k**) a six-gene liver metastatic signature, and (**l**) a five-gene lung metastatic signature were identified. * *p*-value ≤ 0.05; ** *p*-value ≤ 0.005; *** *p*-value ≤ 0.001.

**Figure 5 cancers-15-01831-f005:**
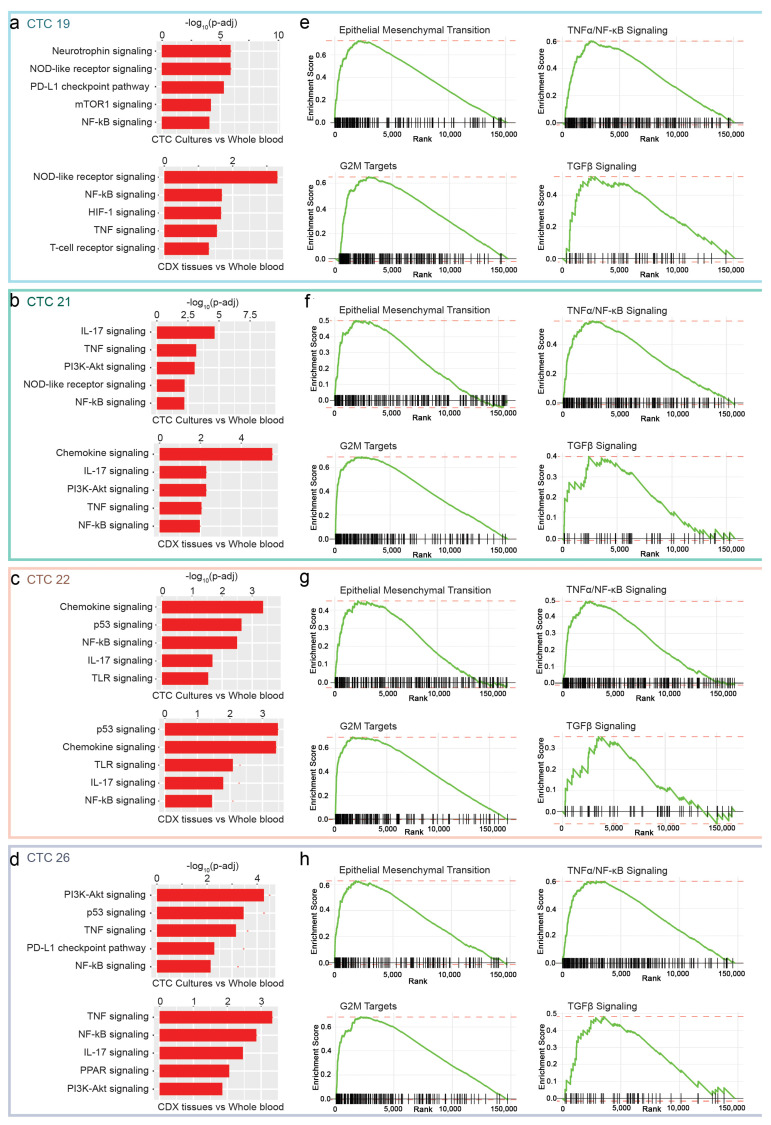
KEGG and GSEA pathway analysis on CTC-derived samples indicate an enrichment of EMT, TGFβ, and NF-κB signaling pathways. Waterfall plots of the top five pathways enriched via KEGG analysis in patient-specific CTC cultures or CDX tissues against whole blood samples for (**a**) patient #19, (**b**) patient #21, (**c**) patient #22, and (**d**) patient #26 are depicted. GSEA enrichment plots depicting enrichment of the hallmark gene sets EMT, TNFα/NF-κB signaling, G2M targets, and TGF-β signaling are shown for samples from patient #19 (**e**), patient #21 (**f**), patient #22 (**g**), and patient #26 (**h**).

**Figure 6 cancers-15-01831-f006:**
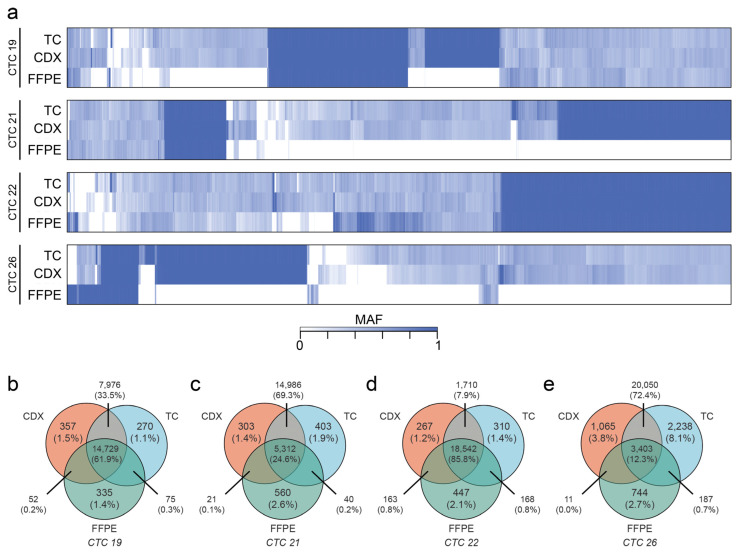
Whole-exome sequencing revealed concordance and discordance between the patient’s tumor and corresponding CTC models (TC and CDX). (**a**) Heat maps depicting the mutant allele fraction (MAF) of all mapped mutations across patient-matched CTC cultures (“TC”), CDX primary tumors (“Tumor”), and clinical formalin-fixed paraffin embedded (“FFPE”) tissue. More mutations were recovered from CTC and CDX tissues than FFPE, on average. An MAF indicates homozygous mutation, while 0.5 indicates heterozygous mutation. (**b**–**e**) Mutations were then mapped in a Venn diagram to determine the number of overlapping and sample-exclusive mutations for each patient.

**Figure 7 cancers-15-01831-f007:**
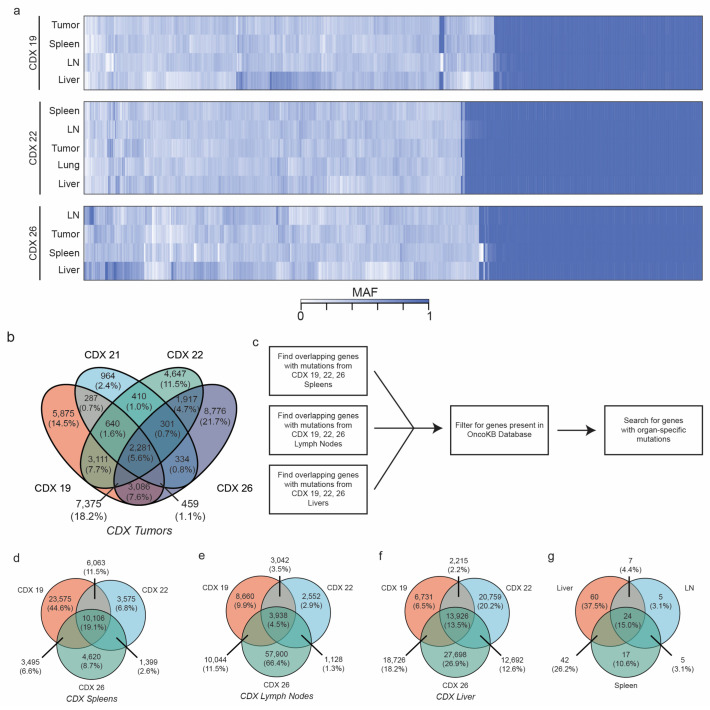
Distant metastases in CDX models were queried to identify organ-specific mutations. (**a**) Heat maps based on patient-specific CDX models indicate that a large proportion of mutations were consistent across CDX primary tumors and distant metastases. (**b**) Mutations across all CDX tumors were mapped to identify the percentage of mutations that were consistent across all CDX tumors. (**c**) A schema for identifying organ-specific mutations is shown. Mutations were mapped to identify the distribution of unique mutations across all CDX (**d**) spleens, (**e**) lymph nodes, and (**f**) livers that had evidence of metastases. (**g**) A Venn diagram evaluating for unique genes with mutations depending on organ evaluated is shown.

**Table 1 cancers-15-01831-t001:** Single-nucleotide polymorphisms were compared between clinical samples of patient primary tumor biopsies and corresponding FFPE, CTC cultures, CDX tumors, and CDX metastatic organ.

Patient	Mutation	Variant Frequency in Biopsy	CDX Metastasis						
			Patient FFPE	CTC Culture	CDX Tumor	Spleen	LN	Liver	Lung
21	APC c.1631T > C	51%	Y *	Y	Y				
	APC c.2521T > G	8%	N **	N	N				
	APC c.2805C > A	34%	Y	N	N				
	APC c.6873A > T	47%	Y	Y	Y				
	PPP2R1A c.548G > A	38%	Y	N	N				
	TP53 c.743G > A	53%	Y	Y	Y				
22	TP53 c.743G > T	48%	Y	Y	N	N	N	N	N
26	SF3B1 c.2098A > G	11%	Y	Y	Y	Y	Y	Y	
	TP53 c.536A > G	28%	Y	N	N	N	N	N	

* Y: Mutation present; ** N: Mutation absent.

## Data Availability

RNA and DNA sequencing data in the current study have been deposited in the Gene Expression Omnibus (GEO) with accession number GSE213141.

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
