# Peer review of "Propagated Circulating Tumor Cells Uncover the Potential Role of NFκB, EMT, and TGFβ Signaling Pathways and COP1 in Metastasis"

_cancers, 2023, doi:10.3390/cancers15061831_

Round 1

Reviewer 1 Report

In this research article, the authors report the establishment and genetic characterization of circulating tumor cell (CTC)-derived models from patients with colon, lung and pancreatic cancer. This was achieved through CTC propagation in vitro and subsequent cell implantation in mice. Furthermore, sequencing studies revealed a five-gene signature of metastatic CTCs.

This study has potential and significant efforts have been put in this study to expand and propagate patient CTCs in aggressive malignancies. Nonetheless, the following major aspects should be addressed to increase its robustness.

Major comments:

-The four paragraphs on page 5 presenting the four patients’ clinical characteristics are too long and dense. The authors may provide instead a patient demographics table showing patient clinical characteristics, available patient samples for sequencing analysis, etc. In addition, a clinical timeline elucidating the different treatments and CTC collection time points may be beneficial for a better understanding.

- The authors claim that 5000 to 25000 cells (CTCs expanded in culture) were injected in mice. What is the estimated number of CTCs (not CTCs plus CD45+) detected initially? I believe it would be important to have this information using for example the CellSearch system.

- IHC staining – in addition to qRT-PCR for epithelial and mesenchymal markers of CTC-derived in vitro cultures- should be performed in comparison to that of CDX tumors (Fig 2d) to confirm the similarities between the established models.

- Regarding qRT-PCR, the primers used should be mentioned. Furthermore, only cytokeratin 18 was used as a CTC marker; the authors should justify why this cytokeratin specifically was used and not other epithelial markers.

- Please clarify what was the sample used for qRT-PCR? Was it cultured CTCs only or CTCs+CD45+? This is important for the interpretation of the results, as no real control is provided for this experiment (except GAPDH).

- It is important to stress on the fact that the “CDX models” herein are not established from patient CTCs directly but from in vitro CTC cultures. The authors compare their success rate and their study findings to CDX establishment attempts directly from patient CTCs (Baccelli et al, Morrow et al, etc) and this may not be the most appropriate control. Nomenclature should be modified: in fact, the authors did not establish CTC-derived xenografts (CDX) but CTC culture-derived xenografts.

- For the identification of somatic mutations, did the authors compare to germline DNA from each patient? If not, this should be discussed.

- In the discussion, please moderate your statement: not only very high CTC counts led to CDX development; for example in the recent scientific article by Tayoun & Faugeroux et al JCI Insight 2022 (PMID 35511434), one CDX was established from a patient with as low as 35 CTCs.

- Finally, it would be beneficial to this research article to perform ex vivo and in vivo pharmacological testings targeting the different enriched pathways.

Minor comments:

- How many healthy donors were included in this study? If there are only two, please replace HD5 and HD6 with HD1 and HD2 to avoid confusion.

- Please list all abbreviations used in a table, it would be much easier to follow. Also make sure all abbreviations used are written in full the first time (for example BWA tool in methods)

- One or more words is/are missing in Methods paragraph 4.10. “As a result, X were generated”. Please complete.   

- In methods or figure legends, please specify the number of experiments or replicates that were performed for sequencing experiments.

Author Response

"Please see the attachment".

Reviewer 2 Report

Here the authors present a novel study involving molecular characterization of circulating tumors from different cancer patients.

The paper is relevant as it describes in details the procedure they used to cultivate the circulating tumor cells, making it feasible to reproduce their approach. 

They validate the tumor specific markers expression profile of the CTCs using histology, as well as compare CTCs versus healthy cells to evaluate transcription heterogeneity and establish a prospective CTC-specific signature.

Pathway analysis on the CTCs then identified altered cellular processes, SNPs and pathways in the system, to identify any potential metastatic markers.

Finally, using exome sequencing they evaluate the genomic similarities between FFPE blocks, primary/ metastatic tumors and patient matched cultured cells.

This paper presents a very interesting pipeline for the field, for a complete characterization of CTCs. The integration of genomics, RNA modulation and histology demonstrate the strengths and weaknesses of culturing the CTCs in a complete manner. The well detailed procedure will enable proper use of the approach described here in the future.

Author Response

"Please see attachment"

Reviewer 3 Report

The manuscript proposed by Jerry Xiao et al provides evidence that CTCs can be efficiently used for ITH studies ex vivo by establishing CDx models. Effective implementation of the methodology and results would pave the way for wide tumor evolution and treatment personalization studies. The paper is well prepared and the results are well elaborated, however, I would suggest some minor remarks before publishing:

1. The methodology description could be trimmed for the main manuscript, and the current full Material and Methods part may be moved into supplementary material.

2. Figure 1 would be more informative to the reader while each cancer type was presented in one row instead of the numerological hierarchy as it is now.

3. The detailed patient description should be moved from the results to the supplementary material

4. The results were obtained by bulk sequencing, while the most recent tumor evolution studies are based on single-cell resolution. Please elaborate deeply on why they did not apply the single-cell sequencing methodology and provide the future perspective of the application of their CDx models establishment with sc-Seqencing/ spatial transcriptomic for tracking the tumor evolution.

Author Response

"Please see attachment".

Round 2

Reviewer 1 Report

In their response to the first reviewing report, the authors estimated that their CTC-CD45+ co-cultures consisted mostly of CTCs (CD45+ are present at <2%) based on their previous study (PMID: 32998338). It is important to consider several aspects here:

-        - Their previous study is based on metastatic breast cancer patients, while the present study characterizes CTCs from patients with colon, lung and pancreatic cancers

-       - In their previous study, the authors used CK 5 and CK 8 as epithelial markers and not CK 18 as used here

Therefore, unfortunately, the comparison with the previous study is not legitimate.

When describing the qRT-PCR results (paragraph 2.1), the authors claim that all CTC cultures co-expressed CK 18 and vimentin, while it is not the case in supplementary figure S1: for example CTC 21 does not express CK. How do the authors explain this? There is no proof that it is the CTCs that are expressing vimentin, since CD45+ cells are known to express vimentin at high levels.

How can they differentiate whether it is CTCs or CD45+ that are being amplified?

Finally, as mentioned in the first report, the authors did not compare the somatic mutations found to germline DNA from each patient, and I believe this constitutes a significant weakness in this paper. A mutation is not considered somatic only because it is found in a cancer mutation panel but because it is not found in the corresponding germline DNA.
